# Clinical Implications of Immune Checkpoints and the RANK/RANK-L Signaling Pathway in High-Grade Canine Mast Cell Tumors

**DOI:** 10.3390/ani13121888

**Published:** 2023-06-06

**Authors:** Noelia C. Talavera Guillén, Andrigo Barboza de Nardi, Felipe Noleto de Paiva, Queila Cristina Dias, Alexandra Pinheiro Fantinatti, Wagner José Fávaro

**Affiliations:** 1Department of Veterinary Clinics and Surgery, São Paulo State University (UNESP), Jaboticabal 14884-900, Brazil; andrigobarboza@yahoo.com.br (A.B.d.N.); n-paiva@hotmail.com (F.N.d.P.); 2Department of Structural and Functional Biology, University of Campinas (UNICAMP), Campinas 13083-970, Brazil; queilaspg@yahoo.com.br (Q.C.D.); alexandrafantinatti@gmail.com (A.P.F.); wjfavaro@gmail.com (W.J.F.)

**Keywords:** mast cell tumor, immune checkpoints, RANK, RANK-L, dogs, immunotherapy

## Abstract

**Simple Summary:**

This study aimed to characterize the molecular profiles of immune checkpoints RANK/RANK-L and IFN-γ in high-grade mast cell tumors and lymph node metastases to understand the complex activities occurring in the tumor microenvironment. All tumors showed moderate or intense immunolabeling of PD-L1, CTLA-4, RANK, RANK-L, and IFN-γ, and the lymph node metastases presented moderate or intense immunolabeling of checkpoint proteins. In conclusion, the high-grade MCTs were characterized as immunosuppressive microenvironments, showing an increase in the RANK/RANK-L signaling pathway and intensified immune checkpoint immunoreactivity, which may explain an intratumoral escape mechanism and indicating high sensitivity to immunotherapy. Therefore, PD-L1, RANK/RANK-L and IFN-γ may be useful in the clinical management of dogs with high-grade MCT.

**Abstract:**

Mast cell tumors (MCTs) are the most common malignant cutaneous tumors in dogs, and they present extremely variable biological behavior. The interaction between RANK, RANK-L, and immune checkpoints is frequently detected in the tumor microenvironment, and, together, they participate in every stage of cancer development. Thus, the aim of this study was to characterize the molecular profiles of PD-L1, CTLA-4, RANK/RANK-L signaling pathway, and IFN-γ in primary tumors and lymph node metastases. Formalin-fixed, paraffin-embedded slides of MCTs and metastatic lymph nodes of ten dogs were submitted to immunohistochemical investigations. The results demonstrated that the tumor microenvironment of the high-grade mast cell tumors showed moderate or intense immunolabeling of all proteins, and the lymph node metastases also showed moderate or intense immunolabeling of checkpoint proteins. In addition, MCTs larger than 3 cm were associated with intensified PD-L1 (*p* = 0.03) in metastatic lymph nodes and RANK-L (*p* = 0.049) immunoreactivity in the tumor. Furthermore, dogs with a survival time of less than 6 months showed higher PD-L1 immunoreactivity (*p* = 0.042). In conclusion, high-grade MCT is associated with an immunosuppressive microenvironment that exhibits elevated RANK/RANK-L signaling and enhanced immune checkpoint immunoreactivity, potentially facilitating intratumorally immune escape. These biomarkers show promise as clinical indicators of disease progression and might response to immunotherapy in dogs with high-grade MCTs, thus emphasizing their importance for guiding treatment decisions and improving outcomes.

## 1. Introduction

Canine mast cell tumors (MCTs) correspond to 16–21% of diagnosed cutaneous neoplasms and are considered the most frequent cutaneous malignant tumors in dogs [1,2]. They are potentially metastatic, affecting primarily regional lymph nodes and, later, liver, spleen, intestine, and, rarely, the lungs. In cases of systemic dissemination, malignant mast cells may infiltrate the bone marrow and peripheral blood [3,4].

The etiology of MCT has not been completely elucidated. However, it may be influenced by chronic inflammation of the skin because their characteristic cytoplasmic granules contain a number of bioactive substances including heparin, histamine, tumor necrosis factor alpha (TNF-α), and several proteases, cytokines, interleukin-6 (IL-6), chemokines (CCL2, CxCL1), growth factors (vascular endothelial growth factor [VEGF], and basic fibroblast factor [bFGF]), and lipid mediators (prostaglandin D_2_ [PGD_2_], and leukotriene C_4_ [LTC_4_]) that are extremely sensitive to chemical degranulation, and are susceptible to exposure to irritating compounds [1,2,3,4,5]. Furthermore, the presence of a mutation in the c-KIT gene (KIT) has been related to tumor development in MCT cases [6,7].

The histopathological classification method devised by Patnaik, Ehler, and MacEwen (1984) [8] revealed a significant relationship between histological grading and patient survival. Considering cell morphology and the extent of tissue involvement, the authors classified MCTs into three defined grades. However, approximately 40% of all the diagnosed MCTs were classified as grade II according to this system, while tumors within this subgroup showed considerable variability in terms of biological behavior. In 2011, Kiupel et al. (2011) [9] proposed a new classification system that divides MCTs into two grades (high and low) based on mitotic count to improve grading objectivity and the ability to prognosticate for dogs with high-grade MCT. Groups and associations of clinical oncologists and pathologists from different countries currently recommend the use of both histological grading systems [2,10]. Nevertheless, any MCT, regardless of grade, can develop aggressive behavior and regional lymph node metastasis, thus worsening the prognosis [1,10].

Due to the heterogeneous behavior of MCT, it is always important to use clinical evaluation, clinical staging, and prognostic markers to anticipate the biological behavior and guide treatment. The overall clinical evaluation must include patient age, clinical progression, tumor size, the site of ulceration, presence of metastasis, clinical stage, and surgical margins, in cases of tumor removal [2,10,11]. Among the prognostic markers, mitotic count, the KIT pattern, c-KIT mutations, and the Ki67 index are considered the most relevant [1,2,12].

Several therapeutic modalities have been currently described, with different success rates, and the choice of one must consider all the aforementioned factors. High-grade MCTs are the most challenging tumors to treat, as they show highly aggressive behavior marked by high metastasis rates, frequent tumor relapses, and rapid disease progression [11,13,14].

Immunotherapy has shown potential as a novel treatment option for cancer, even more so in cases where the treatment outcomes are not always satisfying due to low-to-moderate response rates or limited survival time. Among the available immunotherapy options, therapeutic antibodies targeting immune checkpoint blockage, including programmed cell death 1 (PD-1) and its ligand PD-ligand 1 (PD-L1) and cytotoxic T lymphocyte-associated protein 4 (CTLA-4), have been shown to be a promising approach as anti-cancer treatment by reinvigorating immune responses against cancers [15,16,17]. Regarding the sites where immune checkpoint molecules work, it is now considered that CTLA-4 acts as a negative regulator of the initial activation of T cells in regional lymph nodes, and PD-1 ligands suppress T-cell activation in the tumor microenvironment [18]. However, some studies using animal cancer models showed that, in addition to CTLA-4, PD-1/PD-L1 work as negative regulators in regional lymph nodes, which are the main sites for the induction of antitumor T cells [19,20].

In veterinary medicine, several studies have shown that a variety of canine cancers express PD-L1 and CTLA-4. Ariyarathna et al. (2018) [21] demonstrated that the increased expression of PD-L1 and CTLA-4 was associated with metastasis and poor prognoses in canine mammary gland tumors. The study by Maekawa et al. (2016) [22] revealed the expression of PD-L1 in different tumors, including grade III MCT (classification performed in accordance with the Patnaik grading method).

Another study by the same group of researchers showed that the in vitro blockage of PD-L1 enhanced IFN-γ production by tumor-infiltrating cells, suggesting that anti-PD-L1 antibodies may have therapeutic effects on cancers in dogs [23].

On the other hand, RANK (receptor activator of nuclear factor-κB) and its ligand RANK-L, a member of the TNF-α superfamily, normally are expressed in different types of healthy organs, such as brain, skin, intestine, skeletal muscle, kidney, liver, lung, and mammary tissue, although they are more expressed in bone, lymphoid organs, and the vascular system. However, in the metastatic cascade, the activation of RANK and its ligand increase the survival of circulation metastasis-initiating cancer cells, by stimulating regulatory T cells (Tregs) losing T cell tolerance and protect disseminated cancer cells from immune response [24,25,26].

In human medicine, several studies have shown that the expression of RANK/RANK-L in different types of carcinomas and breast tumors are associated with a higher risk of relapse and death associated with metastases progression [26,27,28,29]. In addition, Chen et al. (2006) [30] studied the expression of RANK-L/RANK/OPG in primary and metastatic human prostate cancer and found that RANK-L/RANK/OPG expression was more frequently observed in skeletal metastases than in lymph node metastases.

In addition, IFN-γ has been shown to play a dual and opposite role in cancer progression. IFN-γ signaling not only enhances PD-L1 expression in tumor cells, inhibiting antitumor immunity, but also increases antigen processing and presentation, thus enhancing their recognition and cytolysis by T cells. A recent study showed that IFN-γ released by effector T cells increased the expression of immunosuppressive markers by tumor-associated lymphatic endothelial cells (LECs). Interestingly, when LECs lacked IFN-γ receptor expression, LEC killing was abrogated, indicating that IFN-γ is indispensable for reducing tumor-associated lymphatic vessel density and drainage [31].

The investigation of checkpoint expression, RANK/RANK-L pathway, and IFN-y is better understood in humans, while in veterinary medicine, these pathways’ study in different neoplasms is still under investigation. The development of new therapeutic strategies, including immunotherapy, has been able to control progression and metastatic dissemination in aggressive neoplasms in humans. The present study aimed to investigate the natural tumor behavior of high-grade MCT in relation to the expression of checkpoints in the tumor and metastases lymph nodes, as well as RANK, RANK-L, and IFN-y in the tumor. The correlation of these factors with clinical information and tumor characteristics was also analyzed to contribute to a better understanding of the aggressiveness of these tumors and the development of new immunotherapy therapeutic options for high-grade MCT.

## 2. Materials and Methods

### 2.1. Case Selection

Ten dogs with MCT from the Oncology Service of the “Governador Laudo Natel” Veterinary Hospital—UNESP—Jaboticabal Campus, were included in the study. MCTs were diagnosed via cytological and histopathological examination. The owners were fully informed of the research content and agreed to participate by signing an informed consent form.

Data on patient history reported by the owners were collected from all dogs, including breed, age, sex, and recurrence, in addition to tumor characteristics: histological grading, based on the classification described by Kiupel et al. (2011) [9], location of the tumor, presence or absence of ulceration, single or multiple nodules, size, and presence of metastasis in the regional lymph node and/or distant organs. The latter two criteria were evaluated via aspiration cytology and abdominal ultrasound investigations, respectively.

All dogs underwent the following laboratory and blood tests: alanine aminotransferase, creatinine, alkaline phosphatase, albumin, total proteins and urinalysis, and imaging tests, including abdominal ultrasound and three-view thoracic X-rays.

The clinical staging of the dogs was performed according to the World Health Organization’s clinical staging system for MCT [2], which considers the number and size of the tumor, the presence of lymph node involvement and distant metastasis, and the presence of systemic signs. No clinical staging was excluded.

The diameter of the tumors was determined using a pachymeter, considering two measurements (length and width), which were categorized as diameters up to 3 cm and diameters greater than 3 cm [2].

To identify the sentinel lymph node, the anatomical location of the primary or recurrent tumor was considered based on the map developed by Suami et al. [32]. On the same day of the surgical procedure, Patent Blue 0.1 mg/kg was applied intradermally around the lymph node. In cases of dogs with MCT in the head or neck, two lymph nodes (mandibular and retropharyngeal) were removed using near-infrared (NIR) imaging [3,4]. The tumor and lymph nodes were stored in 10% neutral buffered formalin solution for histopathological and immunohistochemical analysis. Regarding the assessment of survival time, the patients were monitored clinically at intervals of 3 months for 6 months. After this period, the follow-up was conducted via phone until one year after the end of treatment. The data collected were compiled and organized in tables using Microsoft Excel.

### 2.2. Cytology and Histopathology Analyses

Fine-needle aspiration cytology was performed during the initial consultation to collect cells for cytological examination. A 13 × 4.5 mm fine needle (26 G) was used without aspiration to avoid disrupting the cells. The collected cells were then evaluated using the Romanowski staining technique for diagnosis.

During histopathology investigations surgical excision including lymphadenectomy was performed on all dogs. Only animals with aggressive histomorphology features including a high mitotic index (>7), and at least three multinucleated cells (three or more nuclei) in 10 high-power fields and/or vascular or lymphatic invasion with mast cells were selected for this study.

### 2.3. Immunohistochemical Analyses: PD-L1, RANK, RANK-L, CTLA-4, and IFN-γ

For the immunohistochemical analyses, samples of MCTs and lymph nodes were used. MCT samples were previously fixed, processed, and embedded in paraffin at the Veterinary Pathology Service of UNESP, Jaboticabal, São Paulo, Brazil. Afterward, they were evaluated by two pathologists (histopathological and immunohistochemical investigations).

A Slee CUT5062 RM 2165 rotary microtome (Slee Mainz, Mainz, Germany) was used to cut the samples into 5 μm-thick sections, and antigen retrieval was performed using specific protocols. Next, the sections were incubated in 0.3% H_2_O_2_ to block endogenous peroxidase, and nonspecific binding was blocked by incubating the sections in a blocking solution at room temperature.

The following antibodies were obtained from Santa Cruz Biotechnology Inc. and used for immunohistochemistry: anti-PD-L1 (Pdcd-1L1 (D-8): sc-518027), anti-CTLA-4 (CTLA-4 (F-8): sc-376016), anti-RANK (RANK (H-7): sc-374360), anti-RANK-L (RANK-L (12A668): sc-59925, and anti-IFN-γ (IFN-γRα (GIR-94) sc-12755). All these data are tabulated in Table 1.

Sections were incubated overnight (4 °C) with antibodies diluted at 1% in goat normal serum. Bound antibodies were detected using the EasyLink One Polymer HRP IHC kit (EP-12-20504, EasyPath), following the manufacturer’s instructions, and later stained with diaminobenzidine (DAB) and Harris Hematoxylin. The sections were studied using a Leica DM2500 photomicroscope (Leica, Munich, Germany) equipped with a DFC295 camera (Leica, Munich, Germany).

Mouse urinary bladder tissue sections were utilized as positive controls to evaluate the specificity of both antibodies and protocols employed [33,34]. Furthermore, data from prior studies utilizing cutaneous granuloma from dogs [21] were also utilized. Negative controls included sections of mandibular lymph node, adrenal gland, and pancreas obtained from a dog that died of unrelated causes, as these tissues have been previously demonstrated to not contain PD-L1 protein [21,35].

To evaluate the intensity of antigen immunoreactivity in the tissue samples (MCTs and lymph nodes), ten fields were examined at 400× magnification per dogs and for each antibody (Table 1). The immunolabeling results were analyzed based on the percentage of immunoreactivity through the quantification of immunoreactive/positively-marked cells for each antigen using the ImageJ image analysis program (see Table 2) [35,36,37].

### 2.4. Statistical Analysis

The associations between the immunohistochemical results (percentage of labeling in the nucleus and cytoplasm) and the clinical parameters and MCT characteristics were assessed using the Kruskal–Wallis Test. The results were considered statistically significant when *p* < 0.05, and the software used in the analysis was GraphPad Prism, version 9.0.

## 3. Results

### 3.1. Epidemiological and Clinical Data

The average age of the 10 dogs included in this study was 8.5 years, considering one Pug was only 6 months old. The group of pure breeds was the most representative, corresponding to 60% of the cases, which included Golden Retrievers, Labrador Retrievers, Pugs, Shar-Peis, Pinschers, and Dachshund Terriers. Regarding sex, 60% of the dogs were female (Table 3).

In relation to tumor characteristics, 70% corresponded to tumors larger than 3 cm in diameter. Additionally, the limbs (thoracic and/or pelvic) were the most frequent location, corresponding to 40%, followed by 20% associated with the head and neck region and the thorax, and 10% to the inguinal region and other multiple sites. Single MCT without skin ulcerations were observed in 50% of the cases, whereas single MCT skin ulcerated accounted for 40%, and a single patient had multiple MCTs without ulcerations. Interestingly, 70% of the MCTs were recurrences.

The MCTs measuring more than 3 cm in diameter, which corresponded to 70%, were analyzed based on their characteristics (skin ulcerated or skin non-ulcerated) and the presence or absence of regional and/or distant metastatic lymph nodes (MLNs). Within the skin ulcerated cases, all had MLNs, and only one animal (case No. 4) presented atypical mast cells in the blood. Among the cases of non-ulcerated skin high-grade (60%) MCTs, 83% had regional metastases, while 17% had distant metastases; one of them (case No. 8) had atypical mast cells in the spleen.

Clinical staging was associated with MCT size and survival time, which was corroborated by calling the owners up to one year after treatment. MCTs larger than 3 cm in diameter accounted for 70% of the cases. Dogs staged as IIa accounted for 50% of the cases, followed by 20% staged as IIIa, and 10% staged as Ia, IVa, and IVb. Among all dogs, 50% were still alive. All this information is summarized in Table 4.

### 3.2. Immunolabeling of Proteins PD-L1, CTLA-4, RANK, RANK-L, and IFN-γ

Due to the propensity of high-grade MCT to metastases to lymph nodes, the expression of PD-L1 and CTLA-4 was assessed in both primary tumors and MLN tissues. On the other hand, since RANK, RANK-L, and IFN-γ proteins exhibit limited expression in the MLNs and are predominantly expressed in primary tumors, the present study focuses exclusively on primary tumor tissues for evaluating their expression.

The immunohistochemical examination demonstrated moderate PD-L1 immunoreactivity in the tumors (see Table 5), with an average of 68.9 ± 11.10% and a mean score of 3 (0–4). On the other hand, the MLNs displayed intense immunoreactivity to this protein (as stated in Table 6), with an average of 76.17 ± 7.67% and a mean score of 4 (Figure 1).

CTLA-4 expression in the tumors was also moderate (Table 5), with a mean immunoreactivity of 71.49 ± 8.07% and a mean score of 3 (0–4); however, the immunolabeling of the MLNs showed lower expression of this protein (Table 6), with a mean immunoreactivity of 56.35 ± 11.52% and a mean score of 3 (0–4) (Figure 2).

Regarding the RANK and RANK-L proteins, all MCTs showed moderate cytoplasmic immunoreactivity, with a mean score of 3 (0–4) and a mean of 69.19 ± 5.24% for RANK and 70.50 ± 5.38% for RANK-L (Table 5, Figure 3). Likewise, the expression of IFN-γ in the tumors was moderate, with a mean immunoreactivity of 71.30 ± 7.0% and a mean score of 3 (0–4) (Table 5, Figure 4).

### 3.3. Correlations between Immunoreactivity Associated with Tumor Characteristics and the Clinical Features of the Animals

The associations between the immunohistochemical results (percentage of immunoreactivity to the analyzed proteins) and the clinical parameters and tumor characteristics were evaluated in different scenarios (Table 7 and Table 8).

When analyzing the different characteristics of MCTs, including regional and distant metastases and survival time, no statistical differences were found regarding tumor characteristics and PD-L1 immunoreactivity. However, a statistical difference related to clinical characteristics (*p* = 0.042) was observed, with higher PD-L1 immunoreactivity in dogs surviving for less than 6 months compared to those with longer survival time (Table 7, Figure 5).

The analysis of PD-L1 immunolabeling in the MLNs showed that dogs with MCTs larger than 3 cm in diameter had significantly higher levels of lymph node immunoreactivity to PD-L1 compared to those with smaller MCTs (*p* = 0.03). However, no statistically significant differences were found in relation to clinical characteristics (see Table 8 and Figure 6).

CTLA-4 expression in the MCTs (Table 6, Figure 7) was statistically higher in dogs who had non-ulcerated tumors (*p* = 0.019); however, no statistical differences were observed between the clinical and tumor characteristics and the immunoexpressing of this protein in the MLNs (Table 7).

As for the RANK/RANK-L proteins, a statistical difference (*p* = 0.049) was found when correlating the immunoreactivity of the RANK-L protein in the MCT tissue with tumor size, in which MCTs measuring more than 3 cm showed higher expression of this protein when compared to those smaller than 3 cm (Table 6, Figure 8) In addition, the absence of ulceration in the MCTs showed a statistical difference (*p* = 0.043) regarding the higher expression of the RANK protein compared to the group of MCTs with skin ulcerations (Table 6, Figure 7).

In contrast, no statistically significant differences were found between the MCT and clinical characteristics and IFN-γ protein immunoreactivity (Table 6, Figure 5, Figure 7 and Figure 8).

## 4. Discussion

In the present study, the tumor microenvironment of the high-grade MCTs showed moderate or intense immunolabeling for all proteins (PD-L1, CTLA-4, RANK/RANK-L, and IFN-γ). Moreover, regarding the metastasis lymph nodes (MLNs), all showed intense immunolabeling for PD-L1. According to the tumor characteristics and clinical features of the animals, it was observed that tumors larger than 3 cm presented statistical differences associated with the immunolabeling of PD-L1 in the MLNs and RANK-L in the tumors. Furthermore, we noted that animals with a survival time of less than 6 months showed higher immunoreactivity to PD-L1.

The presence of ulceration in tumors is correlated with mechanisms of chronic inflammation. In the case of mast cell tumors, mast cells could stimulate cells of the innate immune system (macrophages and neutrophils). However, the presence of INF-γ in the tumor microenvironment contributes to the polarization of these cells, which change their function from anti-tumoral to pro-tumoral. These are known as tumor-associated macrophages (TAM) and tumor-associated neutrophils (TAN), respectively. These cells stimulate the production of more pro-angiogenic factors (VEGF-A and FGF-2), and in turn, the release of hypoxia-inducible factor (HIF-α) stimulates further angiogenesis. However, in the present study, it was observed that non-ulcerated MCTs had higher expression of RANK and CTLA-4, which contribute to the tumor immune cascade by recruiting regulatory T cells (Treg), followed by the silencing of CD8+ T cells associated with the release of CCL22 through mast cells and TAMs, contributing to the suppression of T cells through the expression of PD-L1 on the tumor cell membrane. Together, these factors may modify the tumor microenvironment, providing one possible mechanism of tumor escape and contributing to the aggressive behavior observed macroscopically in patients [26,38,39].

In the literature, this is the first time that PD-L1 and CTLA-4 expression has been associated with both tumors and MLNs in high-grade canine MCTs. Ariyarathna et al. (2020) [21] investigated CTLA-4 and PD-1 expression in breast tumors and correlated the two with clinical presentation and survival time. They found that greater immunolabeling was associated with shorter survival time, suggesting that the proteins PD-1 and CTLA-4 are related to the metastatic process. Despite the lack of clarity regarding how checkpoint proteins can interfere in the mechanism of MCT, as well as in the development of metastases, chronic inflammation is considered to be one of the etiologies of MCT, and PD-1/PD-L1 expression by both proinflammatory and neoplastic cells is viewed as a hallmark for T lymphocyte exhaustion [38].

Mast cells are a type of pro-inflammatory cells that are present in all inflammatory processes, including tumor microenvironments. These cells contribute to the process of metastasis in various solid tumors, such as lung cancer [6], renal carcinoma [40], and thyroid tumors, by activating the KIT signaling pathway and its downstream pathways (MAPK and P13K), which promote cell proliferation and survival [41]. Adrenomedullin (AM) expression induced by mast cells facilitates recruitment of endothelial cells to the tumor microenvironment, where they promote angiogenesis via secretion of VEGF, FGF-2, tryptase, and MMPa [30].

A study by Yano et al. (1999) [39] found that the number of mast cells correlated significantly with the depth of invasion, lymph node metastasis, lymphatic or vessel invasion, and histological stage in gastric cancer, and based on it, the authors hypothesized that the release of granular components, such as heparin, histamine, proteases, cytokines, interleukins, and growth factors, might potentiate endothelial cell migration, leading to increased tumor angiogenesis and thereby facilitating MCT progression and aggressiveness behavior [42].

It is worth noting that high expression levels of immune checkpoints, including PD-1, its ligand, and CTLA-4, were observed in neoplastic mast cells, despite limited knowledge on the immunological behavior of mast cell tumors (MCTs). These findings suggest that checkpoint expression, in addition to the c-KIT mutation and downstream pathway activation (MAPK, JAK/STAT3, and PI3K), may be involved in the pathogenesis of MCT. Checkpoints, such as PD-1 and PD-L1, can silence cytotoxic T cells, leading to the suppression of CD8+ T cells through the binding of PD-L1 with the PD-1 receptor expressed by T cells. As a result, there is an increase in regulatory T cells (Tregs) that promote immune evasion and contribute to MCT progression, survival, and migration [41,43,44].

In human neoplasms, PD-L1 expression is present in mammary tumors [45], pancreatic tumors [46], urothelial carcinomas [47], and cutaneous melanomas [48]. PD-1 expression in the tumor is related to a high risk of rapid progression and death, indicating that PD-L1 expression may be associated with poor prognoses in patients with neoplasms [49].

In veterinary medicine, there are a few studies related to the expression of checkpoint proteins in the literature. One study by Maekawa et al. (2016) [22] examined the in vitro expression of PD-1 in different types of cell cultures and found that, among the tumor cells under study, high-grade MCTs showed moderate expression. In another study, Tagawa et al. (2018) [50] investigated the expression of checkpoint proteins in dogs with high-grade B-cell lymphoma, demonstrating that dogs with the disease had higher PD-L1 and CTLA-4 expression compared to healthy ones from the control group. However, the role of CTLA-4 as a prognostic factor within oncology remains controversial; nevertheless, high immunolabeling of this marker is related to a worse prognosis. Additionally, it is known that CTLA-4 expression is associated with the silencing of T cells by different mechanisms. It has even been reported that regulatory T cells (Treg) also express this protein, but in small amounts. Interestingly, CTLA-4 could stimulate the migration of T helper cells to other lymphoid organs [9,21].

The impact of CTLA-4 and PD-1 blockers drastically changes the treatment outcomes of advanced human cancers. Many drugs targeting CTLA-4/PD-1 have been approved for the treatment of different types of cancers, including melanoma, lung, breast, bladder, and gastric cancer, classic Hodgkin’s lymphoma, and B-cell lymphoma [51]. Nevertheless, only a fraction of patients responds to monotherapy; the combination of CTLA-4 and PD-1 blockers showed a remarkable increase in response rates and median survival time in melanoma and renal cell carcinoma [52,53,54]. In dogs, Tagawa et al. (2018) [51] reported high expression of CTLA-4 in high-grade B lymphomas, and similar results were observed in cases of Hodgkin’s and non-Hodgkin’s lymphomas and leukemias in humans [51].

Although the expression of CTLA-4 promotes tumor escape, inhibiting the antitumor response, it has been observed that expression is higher in peripheral blood cells and bone marrow and lower in lymph nodes, contrasting with the results found herein. We can infer that, in addition to low CTLA-4 immunostaining in the lymph nodes and tumors, the presence of neoplastic mast cells in distant organs and peripheral blood, corresponding to cases No. 8 and No. 4, respectively, may be related to the promotion of tumor escape to other organs. Therefore, CTLA-4 expression may not be associated with the prognosis, thus requiring additional studies to understand the relationship with MCT, especially in lymph nodes.

The association of the infiltration of immune and non-immune cells in the tumor microenvironment has allowed researchers to determine the therapeutic response of solid tumors to immunotherapy [9,10], including melanoma. PD-1/PD-L1 expression can guide the possible therapeutic response, as in the case of melanomas that respond to anti-PD-1 therapy, which often correlates with the degree of infiltrating T cells, including CD8+ T cells. Even though the degree of infiltrating T cells is often correlated, one of the causes of resistance to this type of therapy is the presence of mast cells in the tumor microenvironment. Somasundaram et al. (2017) [55] observed that in rats with melanoma, tumor infiltration of mast cells was related to resistance to anti-PD-1 therapy, which raises a new question: could one of the mechanisms of chemoresistance to the treatments already developed for high-grade MCTs be related to high checkpoint expression? In order to answer this question, the tumor microenvironment of MCT needs to be further studied.

Interestingly, in the present study, all high-grade MCTs had a score of 3 in relation to IFN-γ expression. Despite the lack of statistical differences regarding the tumor and clinical characteristics, this could be associated with the higher infiltration of lymphocytes in the tumor (TILs), which may contribute to tumor evasion by promoting tumorigenesis and angiogenesis [56].

The clinical importance of IFN-γ expression in cancer was described by Higgs et al. (2018) [57], who found that patients with small-cell lung carcinoma and advanced-stage urothelial carcinoma had a better response to checkpoint inhibitors (PD-L1), indicating that IFN-γ expression may be considered a predictive marker of response to checkpoint-blocking immunotherapies [58]. IFN-γ induces the expression of CTLA-4 and PD-L1 in tumor cells. The relationship between IFN-γ and CTLA-4 expression in humans has been reported in oral melanoma patients, with CTLA-4 inhibitors (ipilimumab) increasing the response of patients who present higher expression [59]. In this sense, the high expression of CTLA-4 in non-ulcerated tumors could be related to the high expression of IFN-γ.

Under normal conditions, IFN-γ induces PD-L1 expression in antigen-presenting cells and other T-cell-activating cells to prevent tissue damage; however, within the tumor microenvironment, PD-L1 expression is used as an escape strategy by tumor cells. Oyer et al. (2018) [60] demonstrated that increased PD-L1 expression by the tumor generates resistance to Natural Killer (NK) cells and, hence, at the time of IFN-γ/JAK signaling blockage, the NK cells could be reactivated.

These findings have also been observed in gastric carcinomas. PD-L1 expression showed an important relationship with IFN-γ expression; thus, patients with high IFN-γ expression may respond better to PD-L1 inhibitors [61]. Larger tumor sizes in high-grade MCTs may be related to the greater infiltration of T cells into the tumor and higher IFN-γ expression, and, since high-grade MCTs express PD-L1, checkpoint inhibitor treatments might promote a better response with the modulation of IFN-γ.

The association of moderate immunolabeling of RANK-L in high-grade MCTs, especially in tumors with aggressive characteristics (larger than 3 cm), is interesting since it is known that RANK-L is present in more than one phase of metastasis development, including the activation of Treg circulation and facilitating escape from immunosurveillance [12]. Physiologically, this protein is present in several tissues (lymphoid, respiratory, and mammary gland) and in smaller proportions in hematopoietic cells and the spleen. However, it contributes to the activation of T cells and APCs [28,62].

Among the tumor characteristics which correlate with aggressiveness factors are the presence or absence of ulcerations in high-grade MCTs; however, our results showed that non-ulcerated tumors exhibited a higher expression of RANK and CTLA-4, which could provide an explanation for the mechanism of escape of metastases risk. We consider this data crucial, as it could explain how the mechanism of metastasis may already be progressing before the animal presents clinical features of malignancy in cases of high-grade MCTs.

Associating this information with the function of IFN-γ, as mentioned above, this protein can promote an anti-tumor effect. Nonetheless, it can also contribute to tumor development and even help in the development of lymphatic endothelial cells for the progression of lymphatic metastasis, as reported in Chen’s work in animal model studies [30]. The presence of molecules in the tumor microenvironment of high-grade MCTs could potentially activate immunosuppressive pathways, such as the RANK/RANK-L pathways, and promote the development of tumor lymphatic pathways due to high levels of intratumoral IFN-γ. While no statistical differences were found between animal and tumor characteristics, inhibitory therapies targeting these molecules or modulating the function of IFN-y could offer a promising treatment alternative, especially for high-grade MCTs. Alongside the previously mentioned checkpoint inhibitors, such therapies could be considered to enhance treatment outcomes.

In human cancer patients, various tumor types capable of inducing pathological osteolysis have been associated with RANK-L expression, including osteosarcoma, prostatic carcinoma, breast carcinoma, multiple myeloma, and squamous cell carcinoma. Unlike in humans, in veterinary medicine, few studies correlate the expression of RANK and RANK-L in different tumors. Barger et al. (2007) [63] studied the expression of RANK/RANK-L in bone tumors and correlated pain with such expression.

Although the natural behavior of MCT does not involve bone tissue, it is possible to attribute the aggressiveness of the disease to the low therapeutic response to RANK-L expression, which helps tumor cells go unrecognized by the host’s immune system. Therefore, the present study may contribute to the search for new immunotherapy options, as in the case of metastatic breast cancer and melanoma. In Ahern et al. (2018) [64], the authors showed that the combination of PD-1/PD-L1 and CTLA-4 inhibitors in association with RANK-L inhibitors improved the therapeutic response in advanced melanoma patients.

In another study, Galluzzi et al. [62] used RANK/RANK-L blockers in metastatic mammary tumors and observed that their use decreased the carcinogenesis of the tumors and consequently reduced the percentage of metastases in the study group. In the case of MCTs, little is known regarding the mechanism of action and the role of RANK/RANK-L signaling in regional and distant metastases. The high expression of RANK and RANK-L in the tumor cells in the present study and its relationship with tumor size and characteristics may be related to greater aggressiveness, and consequently, therapeutic resistance. Therefore, we recommend more studies addressing the RANK/RANK-L pathway since it could be considered a new target in the development of immunotherapies for MCTs.

In Brazil, a nanoimmunotherapy was developed, known as OncoTherad, which acts as a biological response modifier, triggering stimulation of the Toll-like 4 (TLR4) non-canonical pathways, increasing the expression of TLR4, TRIF, IRF, and IFN-γ [63,64,65]. Reis et al. [34] demonstrated in a chemically induced bladder cancer animal model that OncoTherad reduced RANK/RANK-L protein levels, resulting in decreased PD-1/PD-L1 immunoreactivity, with consequent inhibition of tumor progression. In patients with BCG (Bacillus Calmette–Guerin)-unresponsive non-muscle-invasive bladder cancer, OncoTherad immunotherapy decreased RANK/RANK-L expression, resulting in reduced regulatory T (Treg) cells [34]. In veterinary medicine, OncoTherad immunotherapy has already shown promising results in the treatment of urothelial carcinoma [66] and oral melanoma [67] and may now be considered a novel therapeutic option for high-grade MCTs that can be used in conjunction with other therapies, such as chemotherapy or tyrosine kinase inhibitors (TKI).

## 5. Conclusions

Overall, it is indicated by our findings that high-grade MCT is associated with an immunosuppressive microenvironment that exhibits elevated RANK/RANK-L signaling and enhanced immune checkpoint immunoreactivity, potentially facilitating intratumorally immune escape. These biomarkers hold promise as clinically relevant indicators of disease progression and response to immunotherapy in dogs with high-grade MCTs, emphasizing their importance for guiding treatment decisions and improving patient outcomes.

## Figures and Tables

**Figure 1 animals-13-01888-f001:**
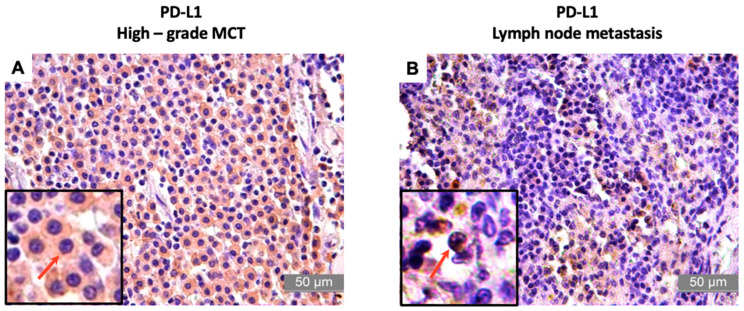
Photomicrographs of the immunostaining for PD-L1 in high-grade canine MCT and lymph node metastasis. (**A**,**B**): The arrows indicate showing cell membrane and cytoplasm staining positive for PD-L1 in MCT and lymph node metastasis, characterized by the brown coloration, which is shown by diaminobenzidine (DAB) staining and hematoxylin counterstaining. Bar = 50 µm.

**Figure 2 animals-13-01888-f002:**
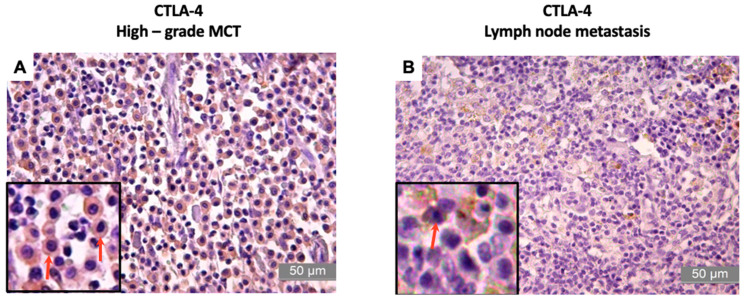
Photomicrographs of the immunostaining for CTLA-4 in high-grade canine MCT and lymph node metastasis. (**A**,**B**): The arrows indicate showing cell membrane and cytoplasm staining positive for CTLA-4 in MCT and lymph node metastasis characterized by the brown coloration, which is shown by diaminobenzidine (DAB) staining and hematoxylin counterstaining. Bar = 50 µm.

**Figure 3 animals-13-01888-f003:**
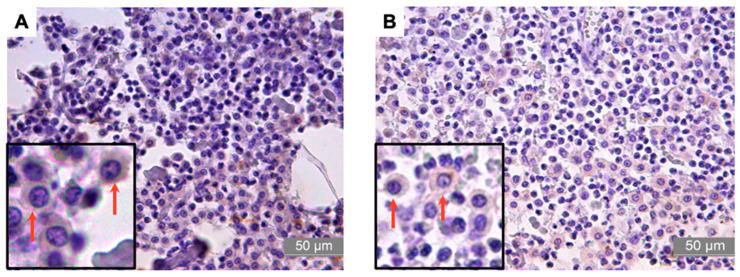
Photomicrographs of the immunostaining for RANK and RANK-L in high-grade canine MCT. (**A**) Photomicrograph of the immunostaining for RANK in high-grade canine MCT, The arrows indicate showing cell membrane and cytoplasm staining positive for RANK. (**B**) Photomicrograph of the immunostaining for RANK-L in high-grade canine MCT, The arrows indicate showing cell membrane and cytoplasm staining positive for RANK-L. characterized by the brown coloration, which is shown by diaminobenzidine (DAB) staining and hematoxylin counterstaining. Bar = 50 µm.

**Figure 4 animals-13-01888-f004:**
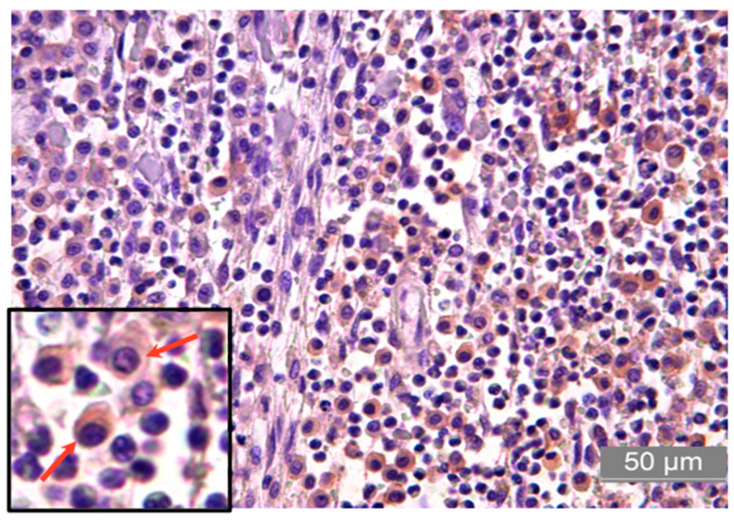
Photomicrograph of the immunostaining for IFN-γ in high-grade canine MCT. The arrows indicate showing cell membrane and cytoplasm staining positive for IFN-γ positive immunoreactivity, characterized by the brown coloration, which is shown by diaminobenzidine (DAB) staining and hematoxylin counterstaining. Bar = 50 µm.

**Figure 5 animals-13-01888-f005:**
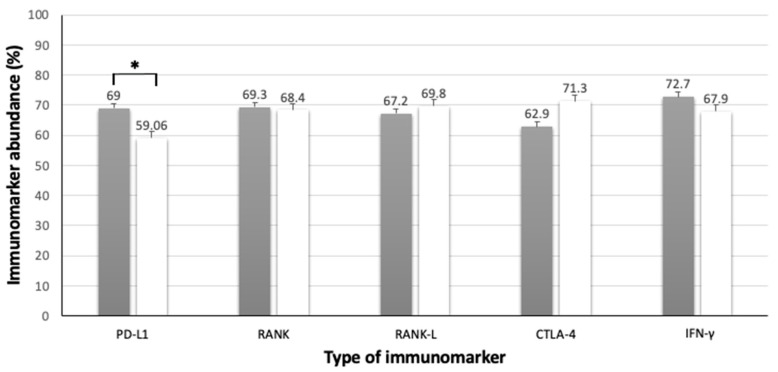
Association between immunolabeling of PD-L1, RANK, RANK-L, CTLA-4 and IFN-γ of high-grade MCTs and survival time. Black and white columns indicate survival time < 6 months and >6 months, respectively. Error bars are indicated in black in each column, together with the mean immunomarker abundance. ***** = statistical difference (*p* = 0.042). Confidence interval 95%. Kruskal–Wallis test.

**Figure 6 animals-13-01888-f006:**
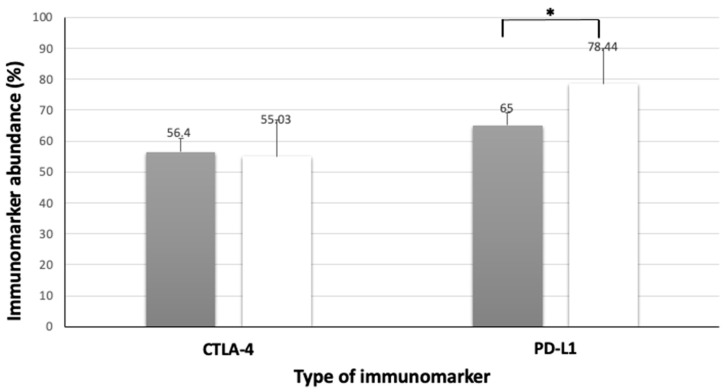
Immunolabeling of PD-L1 and CTLA-4 in metastases lymph nodes associated with high-grade MCT size. Black columns indicate tumors smaller than 3 cm and white columns indicate tumors larger than 3 cm. Error bars are indicated in black in each column, along with the mean immunomarker abundance. ***** = statistical difference (*p* = 0.03). Confidence interval 95%. Kruskal–Wallis test.

**Figure 7 animals-13-01888-f007:**
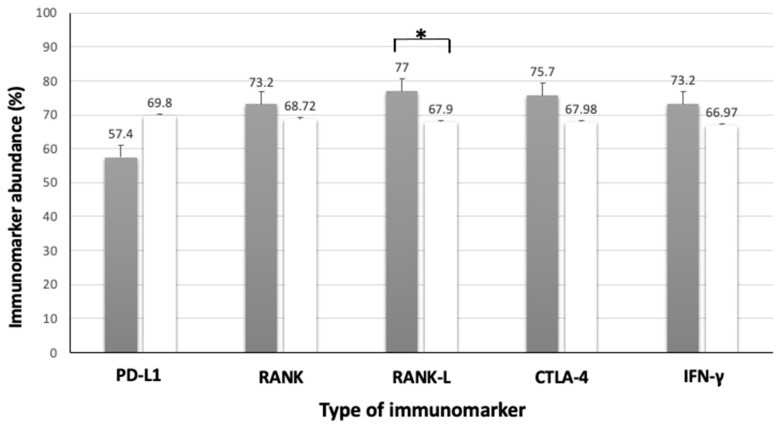
Association between immunolabeling of PD-L1, RANK, RANK-L, CTLA-4 and IFN-γ and high-grade MCT size. Black columns indicate high-grade MCTs larger than 3 cm and white columns indicate high-grade MCTs smaller than 3 cm. Error bars are indicated in black in each column, along with the mean immunomarker abundance. ***** = statistical difference (*p* = 0.049). Confidence interval 95%. Kruskal–Wallis test.

**Figure 8 animals-13-01888-f008:**
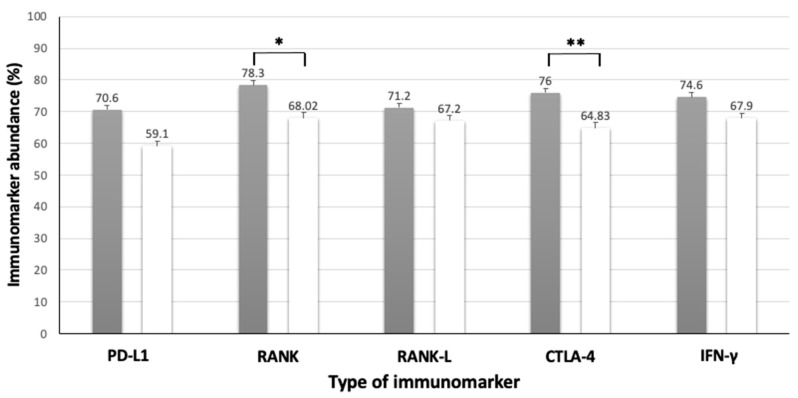
Association between presence or absence of high-grade MCT skin ulceration and the immunolabeling of PD-L1, RANK, RANK-L, CTLA-4, and IFN-γ. Black columns indicate MCT skin non-ulceration and white bars indicate presence of MCT skin ulceration. Error bars are indicated in black in each column, together with the mean immunomarker abundance. * = statistical difference (*p* = 0.043), ** = statistical difference (*p* = 0.019). Confidence interval 95%. Kruskal–Wallis test.

**Table 1 animals-13-01888-t001:** Antibodies tested and antibodies dilution used for MCT investigation.

Antibody	Code	Distributor	Dilution
Mouse, monoclonal anti-PD-L1	sc-518027	*DAKO*, *USA*	MCT skin: 1:25 MCT lymph node: 1:100
Mouse, monoclonal anti-CTLA-4	sc-376016	*Santa Cruz Biotechnology*, *USA*	MCT skin: 1:50 MCT lymph node: 1:100
Mouse, monoclonal anti-RANK	sc-374360	*Santa Cruz Biotechnology*, *USA*	MCT skin: 1:50
Mouse, monoclonal anti- RANK-L	sc-52950	*Santa Cruz Biotechnology*, *USA*	MCT skin: 1:50
Rabbit, polyclonal anti-IFN-γ Rα	sc-12755	*Santa Cruz Biotechnology*, *USA*	MCT skin: 1:50

**Table 2 animals-13-01888-t002:** Scoring system for immunoreactivity in tissue samples of MCT and metastases lymph nodes.

Score	Percentage Cell Immunoreactivity	Interpretation
0	0%	No immunoreactivity
1	1–25%	Poor immunoreactivity
2	25.1–50%	Mild immunoreactivity
3	50.1–75%	Moderate immunoreactivity
4	More than 75%	Intense immunoreactivity

**Table 3 animals-13-01888-t003:** Dogs recruited in the study: breed, age and sex.

Dog No.	Breed	Age	Sex
**1**	Golden Retriever	9 years	Female
**2**	Pug	0.6 years	Female
**3**	Shar-Pei	6.6 years	Male
**4**	Mongrel	15 years	Female
**5**	Dachshund	8 years	Female
**6**	Mongrel	6 years	Female
**7**	Boxer	7 years	Male
**8**	Mongrel	13 years	Male
**9**	Labrador Retriever	7 years	Male
**10**	Mongrel	10 years	Female

**Table 4 animals-13-01888-t004:** Dogs included in the study: clinical characteristics.

Patient No.	MCT Size	Skin Regional Location	Skin Ulceration	Stage	Recurrence	Metastasis	Survival
**1**	>3 cm	Pelvic limb	Yes	IIa	Yes	Lymph node	>6 months
**2**	<3 cm	Ear	No	IIa	Yes	Lymph node	>6 months
**3**	>3 cm	Multiple	No	IIIa	Yes	Lymph node	>6 months
**4**	>3 cm	Thoracic limb	Yes	IVb	Yes	Lymph node and blood	<6 months
**5**	<3 cm	Inguinal	No	IIIa	Yes	Lymph node	>6 months
**6**	>3 cm	Thorax	No	IIa	No	Lymph node	>6 months
**7**	>3 cm	Base of ear	Yes	IIa	Yes	Lymph node	<6 months
**8**	>3 cm	Pelvic limb	No	Iva	No	Spleen and lymph node	<6 months
**9**	<3 cm	Thorax	No	Ia	No	No	>6 months
**10**	>3 cm	Thoracic limb	Yes	IIa	Yes	Lymph node	<6 months

**Table 5 animals-13-01888-t005:** Score and percentage of immunoreactivity to the different antigens in high-grade canine MCT.

PD-L1	CTLA-4	RANK	RANK-L	IFN-γ
3	3	3	3	3
(68.9 ± 11.10%)	(71.49 ± 8.07%)	(69.19 ± 5.24%)	(70.50 ± 5.38%)	(71.30 ± 7%)

Scores (1–4) correspond to the intensity of immunoreactivity to each protein (1 = very weak, 2 = weak, 3 = moderate, and 4 = intense). Values between parentheses indicate the means ± standard deviations of the percentage of cells positive to the antigens PD-L1, CTLA-4, RANK, RANK-L, and IFN-γ (*n* = 10 sections/patient).

**Table 6 animals-13-01888-t006:** Score and percentage of immunoreactivity to the antigens PD-L1 and CTLA-4 in the lymph node metastases.

PD-L1	CTLA-4
4	3
(76.17 ± 7.67%)	(56.35 ± 11.52%)

Scores (1–4) correspond to the intensity of immunoreactivity to each protein (1 = very weak, 2 = weak, 3 = moderate, and 4 = intense). Values between parentheses indicate the means ± standard deviations of the percentage of cells positive to the antigens PD-L1 and CTLA-4 (*n* = 10 sections/patient).

**Table 7 animals-13-01888-t007:** Score and percentage of immunoreactivity to the different antigens in high-grade canine MCT and/or clinical characteristics.

High-Grade Canine MCT Immunoreactivity						
		PD-L1	CTLA-4	RANK	RANK-L	IFN-y
**Survival time**	**<6 months**	3	3	3	3	3
		(59.06 ± 11.52%) *	(71.3 ± 5.15)	(68.4 ± 5.49%)	(69.8 ± 3.98%)	(67.9 ± 6.81%)
	**>6 months**	3	3	3	3	3
		(69.0 ± 6.85%)	(62.9 ± 11.84%)	(69.3 ± 5.63%)	(67.2 ± 8.35%)	(72.7 ± 7.88)
**Tumor size**	**>3 cm**	3	3	3	4	3
		(69.8 ± 10.78%)	(67.98 ± 7.45%)	(68.7 ± 1.96%)	(77.0 ± 5.96%) *	(66.97 ± 6.40%)
	**<3 cm**	3	4	3	3	3
		(57.4 ± 14.79%)	(75.7 ± 5.49%)	(73.2 ± 4.63%)	(67.9 ± 3.68%)	(73.2 ± 7.88%)
**Characteristics**	**Skin ulcerated**	3	3	3	3	3
		(59.1 ± 13.95%)	(64.83 ± 7.39%)	(68.02 ± 1.96%)	(67.2 ± 3.98)	(67.9 ± 6.49)
	**Skin Non-ulcerated**	3	4	4	3	3
		(70.6 ± 2.99%)	(76.0 ± 4.61%) *	(78.32 ± 4.63) *	(71.23 ± 8.35%)	(74.6 ± 7.01)

Scores (1–4) correspond to the intensity of immunoreactivity to each protein (1 = very weak, 2 = weak, 3 = moderate, and 4 = intense). Values between parentheses indicate the means ± standard deviations of the percentage of cells positive to the antigens PD-L1, CTLA-4, RANK, RANK-L, and IFN-γ (*n* = 10 sections/patient). * = statistical difference (*p* < 0.05) according to the Kruskal–Wallis test.

**Table 8 animals-13-01888-t008:** Score and percentage of immunoreactivity to the different antigens in the metastatic lymph nodes and/or clinical characteristics.

Immunoreactivity of the Different Antigens in Metastatic Lymph Nodes			
		PD-L1	CTLA-4
**Status**	**Survival time< 6 months**	3	3
		(69.70 ± 8.26%)	(56.35 ± 11.01%)
	**Survival time > 6 months**	4	3
		(78.44 ± 5.69%)	(61.18 ± 11.23%)
**Characteristics**	**MCT skin Ulcerated**	3	3
		(78.44 ± 6.09%)	(55.03 ± 11.48%)
	**MCT skin Non-ulcerated**	3	3
		(70.9 ± 8.22%)	(56.35 ± 10.55%)
**MCT size**	**>3 cm**	4	3
		(78.44 ± 5.57%) *	(55.03 ± 11.03%)
	**<3 cm**	3	3
		(65.00 ± 4.06%)	(56.4 ± 12.01%)

Scores (1–4) correspond to the intensity of immunoreactivity to each protein (1 = very weak, 2 = weak, 3 = moderate, and 4 = intense). Values between parentheses indicate the means ± standard deviations of the percentage of cells positive to the antigens PD-L1 and CTLA-4 (*n* = 10 sections/patient). * = statistical difference (*p* < 0.05) according to the Kruskal–Wallis test.

## Data Availability

The data presented in this study are available upon request from the corresponding author.

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
