# Peer review of "Clinical Implications of Immune Checkpoints and the RANK/RANK-L Signaling Pathway in High-Grade Canine Mast Cell Tumors"

_animals, 2023, doi:10.3390/ani13121888_

Round 1
Reviewer 1 Report
1. The bar 50 μm in figure 2B is not in the same place as other figures, please revise.
2. Figure 5-8 are not clear, please revise.
3. The authors should explain “An interesting finding was the expression of RANK and CTLA-4 in non-ulcerated tumors. According to MCT prognostic factors, the presence of ulceration is associated with a worse prognosis; however, the expression of these proteins could provide insight into how high-grade MCT promote an intratumoral escape mechanism.” in detailed.
4. Further, the authors should explain “It is noteworthy that, despite the lack of knowledge on the immunological behavior of MCT, we observed high checkpoint expression (PD-1 and its ligand and CTLA-4), indicating that, in addition to the c-KIT mutation, within the pathogenesis of MCT, neoplastic mast cells express checkpoints, which enable the silencing of cytotoxic T cells based on the PD-1/PD-L1 axis. The cell surface expression of PD-L1 is carried out by antigen-presenting dendritic cells (APCs), which require the binding of the type II histocompatibility complex (MHCII) to CD4+ T lymphocytes, and, consequently, convert them into effector T lymphocytes (CD4+)” in detailed.1. The bar 50 μm in figure 2B is not in the same place as other figures, please revise.
2. Figure 5-8 are not clear, please revise.
3. The authors should explain “An interesting finding was the expression of RANK and CTLA-4 in non-ulcerated tumors. According to MCT prognostic factors, the presence of ulceration is associated with a worse prognosis; however, the expression of these proteins could provide insight into how high-grade MCT promote an intratumoral escape mechanism.” in detailed.
4. Further, the authors should also explain “It is noteworthy that, despite the lack of knowledge on the immunological behavior of MCT, we observed high checkpoint expression (PD-1 and its ligand and CTLA-4), indicating that, in addition to the c-KIT mutation, within the pathogenesis of MCT, neoplastic mast cells express checkpoints, which enable the silencing of cytotoxic T cells based on the PD-1/PD-L1 axis. The cell surface expression of PD-L1 is carried out by antigen-presenting dendritic cells (APCs), which require the binding of the type II histocompatibility complex (MHCII) to CD4+ T lymphocytes, and, consequently, convert them into effector T lymphocytes (CD4+)” in detailed.
Author Response
The authors should explain “An interesting finding was the expression of RANK and CTLA-4 in non-ulcerated tumors. According to MCT prognostic factors, the presence of ulceration is associated with a worse prognosis; however, the expression of these proteins could provide insight into how high-grade MCT promote an intratumoral escape mechanism.” in detailed.
The presence of ulceration in tumors is correlated with mechanisms of chronic inflammation. In the case of mast cell tumors, mast cells could stimulate cells of the innate immune system (macrophages and neutrophils). However, the presence of INF-γ in the tumor microenvironment contributes to the polarization of these cells, which change their function from anti-tumoral to pro-tumoral. These are known as tumor-associated macrophages (TAM) and tumor-associated neutrophils (TAN), respectively. These cells stimulate the production of more pro-angiogenic factors (VEGF-A and FGF-2), and in turn, the release of hypoxia-inducible factor (HIF-α) stimulates further angiogenesis. However, in the present study, it was observed that non-ulcerated MCTs had higher expression of RANK and CTLA-4, which contribute to the tumor immune cascade by recruiting regulatory T cells (Treg), followed by the silencing of CD8+ T cells associated with the release of CCL22 through mast cells and TAMs, contributing to the suppression of T cells through the expression of PD-L1 on the tumor cell membrane. Together, these factors may modify the tumor microenvironment, providing one possible mechanism of tumor escape and contributing to the aggressive behavior observed macroscopically in patients [26, 42, 64].The bar 50 µm in figure 2B is not in the same place as other figures, please revise.
Figure 5 - 8 are not clear, please revise: Here we improve all the figures, but we would like to know more specifications to can improve:
Figure 5: Association between immunolabeling of PD-L1, RANK, RANK-L, CTLA-4 and IFN-γ of high-grade MCT tumors and survival time. Black and white columns indicate survival time < 6 months and > 6 months, respectively. Error bars are indicated in black in each column, together with the mean immunomarker abundance. * = statistical difference (p=0.042). Confidence interval 95%. Kruskall-Wallis test.
Figure 6: Immunolabeling of PD-L1 and CTLA-4 in metastatase lymph nodes associted with MCT high-grade tumor size. Black columns indicate tumors smaller than 3 cm and white columns indicate tumors larger than 3 cm. Error bars are indicated in black in each column, along with the mean immunomarker abundance. * = statistical difference (p=0.03). Confidence interval 95%. Kruskall-Wallis test.
Figure 7: Association between immunolabeling of PD-L1, RANK, RANK-L, CTLA-4 and IFN-γ and MCT high-grade tumor size. Black columns indicate MCT high-grade tumors larger than 3 cm and white columns indicate MCT high-grade tumors smaller than 3 cm. Error bars are indicated in black in each column, along with the mean immunomarker abundance. * = statistical difference (p=0.049). Confidence interval 95%. Kruskall-Wallis test.
Figure 8: Association between presence or absence of MCT- high grade skin ulceration and the immunolabeling of PD-L1, RANK, RANK-L, CTLA-4, and IFN-γ. Black columns indicate MCT skin non-ulceration and white bars indicate presence of MCT skin ulceration. Error bars are indicated in black in each column, together with the mean immunomarker abundance. * = statistical difference (p=0.043), ** = statistical difference (p=0.019). Confidence interval 95%. Kruskall-Wallis test.

Reviewer 2 Report
Dear Authors, the paper is original and could have interest in readers for its good scientific soundness. The article needs a major revision.
The indications for making improvements are indicated below as attention points or by line/s.
Attention points
a) In writing the paper it is recommended to use impersonal verb forms;
b) When possible use the word "dogs" instead "patients" for devoiding confusion about species (human or canine);
c) when possible use "MCT" instead "tumor" for improving the effectiveness of communication.
d) some part of article needs a linguistic revision and it is necessary re-phrase the sentence (the part of rephrasing will be indicated by line/s).
By line
Please, operate the following revision
Line 17 - detail the wording "of all proteins";
Line 22 - detail the wording "biomarkers";
Line 30 - change "samples" with "slides";
Line 31 - change "animals " with "dogs";
Line 32 - change "analysis" with "investigations";
Line 32 - "Our results..." re-phrase the sentence in impersonal verbal form;
Line 42 - cancel "the" before the word liver and lungs;
Line 68 - Change words order "important always" in "always important";
Lines 103-104 "tissues, such as brain...". Brain is an organ not a tissues (nervous tissue) and so on for the other organs cited in the sentence. Please change tissue in organs or organs in tissue. You can choice one or the other verbal solution but in any case you must rephrase the sentence.
Line 109 - Histology classified the physiological tissue. Neoplastic tissue is a conversational form used by pathologists but as tissue do not exits, while neoplasia yes. Please re-phrase the sentence;
Line 115 - Change "development" in "progression";
Line 122 - Change "developed" with "performed";
Lines 120-130 - Re-phrase the sentence for improving communication effectiveness;
Line 135 - Change "this" with "the";
Line 135 - Change "All of them" with "MCT";
Line 135 - The sentence describe the diagnostic approach (cytological and histopathological), but in the Chapter "Materials and Methods" no methodology, for histopathology and even less for cytology, is described. Please include a sub-chapter, before the description of immunohistochemistry, providing their methodological descriptions.
Line 139 - Change "animals" with "dogs";
Line 144 - Include the word "investigations" after the word "ultrasound";
Line 161 - Specify if the formalin was a buffered formalin and eventually in dedicate the value of pH;
Lines 164-165 Re-phrase the sentence improving I.T. lexical appropriateness;
Line 169 - Change "the samples " with "MCT samples";
Line 172 - Change "analysis" with "investigations";
Table 1 change the "Description..." with "Antibodies tested and antibodies dilution used for MCT investigation ";
Table 1 column Dilution improve the communicability changing "Tumor" with "MCT skin" and "Lymph node" with "MCT Lymph node";
Lines 188-189 Re-phrase the sentence as follow "Sections were incubated overnight (4°C) with antibodies diluted al 1% in goat normal serum";
Line 192 - Change "evaluated" with "studied";
Lines 194-197 Re-phrase the sentence;
Lines 198-208 Re-phrase the sentence and summarize in a Table the scoring system;
Table 2 change "Patients" with "Dogs recruited in the study: breed, age and sex" (Please indicate dogs neutered if included in the recruiting)";
Table 3 column "Location" change with "Skin location" and column "Ulceration" with "Skin ulceration";
Lines 249-253 Re-phrase the sentence;
Tables 4-7 Include "Score and" before "percentage";
Line 297 - Change "Similarly" with "Likewise";
Lines 331-336 Re-phrase the sentence;
Figure 5 Include the word "Survival time" before "< 6 months" and "> 6 months";
Line 348 change "patients with tumors" with "dogs with MCT";
Figures 6-7 Include the word "MCT size" before "< 3 cm" and "> 3 cm";
Line 359 - Change "tumor tissue" (which dog not exits) with "MCT";
Figure 8 - Include the word "MCT skin" before "non-ulcerated" and " ulcerated ";
Line 390 - Please use the extended form of MLNs the first time and include the acronyms in round brackets);
Line 400 - Change the incipit "To our knowledge, this " with "In literature ...";
Line 419 - insert the word "risk" after "metastasis". Please support with a citation the MCT-related heparin and istamin that can potentially may favour MCT "metastasis";
Lines 432-433 Delete "among others" because too generic and not supported by citation;
Line 441 - Change "patients" with "dogs";
Line 442 -Change "healthy dogs" with "healthy ones";
Line 510 - Change "pulmonary" with "respiratory";
Line 552 - "We can deduce that, in addition.." Use an impersonal verbal form and re-phrase the sentence;
Conclusion - Re-phrase the sentences for improving the scientific communicability.
The paper needs some language revisions as indicated in analytical way in the box "Comments and Suggestions".
Author Response
Good night:
I have attachment one file with all the corrections including word modifications, this document also include the review form.
Here Im explain line by line:
Line 32 - "Our results..." re-phrase the sentence in impersonal verbal form;
The results demonstrated that the tumor microenvironment of the high-grade mast cell tumors showed moderate or intense immunolabeling of all proteins, and the lymph node metastases also showed moderate or intense immunolabeling of checkpoint proteins.
Lines 103-104 "tissues, such as brain...". Brain is an organ not a tissues (nervous tissue) and so on for the other organs cited in the sentence. Please change tissue in organs or organs in tissue. You can choice one or the other verbal solution but in any case you must rephrase the sentence.
On the other hand, RANK (receptor activator of nuclear factor-κB) and its ligand RANK-L, a member of the TNF-α superfamily, normally are expressed in different types of healthy organs, such as brain, skin, intestine, skeletal muscle, kidney, liver, lung, and mammary tissue, although they are more expressed in bone, lymphoid organs, and the vascular system. However, in the metastatic cascade, the activation of RANK and its ligand increase the survival of circulation metastasis-initiating cancer cells, by stimulating regulatory T cells (Tregs) losing T cell tolerance and protect disseminated cancer cells from immune response [23,24].
Line 109 - Histology classified the physiological tissue. Neoplastic tissue is a conversational form used by pathologists but as tissue do not exits, while neoplasia yes. Please re-phrase the sentence;
In human medicine, several studies have shown that the expression of RANK/RANK-L in different types of carcinomas and breast tumors are associated with a higher risk of relapse and death associated with metastases progression[25-27].
Lines 120-130 - Re-phrase the sentence for improving communication effectiveness;
The investigation of checkpoint expression, RANK/RANK-L pathway, and IFN-y is better understood in humans, while in veterinary medicine, these pathways' study in different neoplasms is still under investigation. The development of new therapeutic strategies, including immunotherapy, has been able to control progression and metastatic dissemination in aggressive neoplasms in humans. The present study aimed to investigate the natural tumor behavior of high-grade MCT in relation to the expression of checkpoints in the tumor and metastases lymph nodes, as well as RANK, RANK-L, and IFN-y in the tumor. The correlation of these factors with clinical information and tumor characteristics was also analyzed to contribute to a better understanding of the aggressiveness of these tumors and the development of new immunotherapy therapeutic options for high-grade MCT.
Line 135 – The sentence describe the diagnostic approach (cytological and histopathological), but in the chapter “materials and methods” no methodology, for histopathology and even less for cytology, is drecribed. Please, include a sub-chapter, before the description of immunohistochemistry, providing their methodological descrptions.
Fine-needle aspiration cytology was performed during the initial consultation to collect cells for cytological examination. A 13 x 4.5 mm fine needle (26 G) was used without aspiration to avoid disrupting the cells. The collected cells were then evaluated using the Romanowski staining technique for diagnosis.
During histopathology analyses, surgical excision including lymphadenectomy was performed on all dogs. Was selected for the study only animals with aggressive histomorphology features including a high mitotic index (>7), At least three multinucleated cells (three or more nuclei) in in 10 high-power fields and/or vascular or lymphatic invasion with mast cells.
Line 194 – 197 Re phrase the sentence:
Mouse urinary bladder tissue sections were utilized as positive controls to evaluate the specificity of both antibodies and protocols employed. Furthermore, data from prior studies utilizing cutaneous granuloma from dogs [33] was also utilized. Negative controls included sections of mandibular lymph node, adrenal gland, and pancreas obtained from a dog that died of unrelated causes, as these tissues have been previously demonstrated to not contain PD-L1 protein [34-36].
Line 161 - Specify if the formalin was a buffered formalin and eventually in dedicate the value of pH;
The tumor and lymph nodes were stored in 10% neutral buffered formalin solution for histopathological and immunohistochemical analysis.
Lines 164-165 Re-phrase the sentence improving I.T. lexical appropriateness
Regarding the assessment of survival time, the patients were monitored clinically at intervals of 3 months for 6 months. After this period, the follow-up was conducted via phone until one year after the end of treatment. The data collected were compiled and organized in tables using Microsoft Excel.
Lines 198-208 Re-phrase the sentence and summarize in a Table the scoring system;
To evaluate the intensity of antigen immunoreactivity in the tissue samples (MCT tumor and lymph nodes), ten fields were examined at 400x magnification per dogs and for each antibody (Table 1). The immunolabeling results were analyzed by percentage of immunoreactivity through the quantification of immunoreactive/positively-marked cells for each antigen by the ImageJ image analysis program (table 2).
|
Score |
Percentage cell immunoreactivity |
Interpretation |
|
0 |
0% |
No immunoreactivity |
|
1 |
1%-25% |
Poor immunoreactivity |
|
2 |
25.1% - 50% |
Mild immunoreactivity |
|
3 |
50.1% - 75% |
Moderate immunoreactivity |
|
4 |
More than 75% |
Intense immunoreactivity |
Lines 249-253 Re-phrase the sentence;
Due to the propensity of high-grade MCT to metastasize to lymph nodes, the expression of PD-L1 and CTLA-4 was assessed in both primary tumors and MLN tissues. On the other hand, since RANK, RANK-L, and IFN-γ proteins exhibit limited expression in the MLNs and are predominantly expressed in primary tumors, the present study focuses exclusively on primary tumor tissues for evaluating their expression.
The immunohistochemical examination demonstrated moderate PD-L1 immunoreactivity in the tumors (see Table 5), with an average of 68.9 ± 11.10% and a mean score of 3 (0-4). On the other hand, the MLNs displayed intense immunoreactivity to this protein (as stated in Table 6), with an average of 76.17 ± 7.67% and a mean score of 4 (Figure 1).
Lines 331-336 Re-phrase the sentence;
When analyzing the different characteristics of MCT, including regional and distant metastases and survival time, no statistical differences were found regarding tumor characteristics and PD-L1 immunoreactivity. However, a statistical difference related to clinical characteristics (p=0.042) was observed, with higher PD-L1 immunoreactivity in dogs surviving for less than 6 months compared to those with longer survival time (Table 6, Figure 5).
Line 419 - insert the word "risk" after "metastasis". Please support with a citation the MCT-related heparin and istamin that can potentially may favour MCT "metastasis";
Mast cells are a type of pro-inflammatory cells that are present in all inflammatory processes, including tumor microenvironments. These cells contribute to the process of metastasis in various solid tumors such as lung cancer [6], renal carcinoma [39] , and thyroid tumors by activating the KIT signaling pathway and its downstream pathways (MAPK and P13K), which promote cell proliferation and survival[40]. Adrenomedullin (AM) expression induced by mast cells facilitates recruitment of endothelial cells to the tumor microenvironment, where they promote angiogenesis via secretion of VEGF, FGF-2, tryptase, and MMPa [30].
A study by Yano et al. (1999) [41] found that the number of mast cells correlated significantly with the depth of invasion, lymph node metastasis, lymphatic or vessel invasion, and histological stage in gastric cancer, based on it, the authors hypothesized that the release of granular components, such as heparin, histamine, proteases, cytokines, interleukins, and growth factors, might potentiate endothelial cell migration, leading to increased tumor angiogenesis and facilitating MCT progression and aggressiveness behavior[42].
Line 552 - "We can deduce that, in addition.." Use an impersonal verbal form and re-phrase the sentence;
The presence of molecules in the tumor microenvironment of high-grade MCT could potentially activate immunosuppressive pathways, such as the RANK/RANK-L pathways, and promote the development of tumor lymphatic pathways due to high levels of intratumoral IFN-γ. While no statistical differences were found between animal and tumor characteristics, inhibitory therapies targeting these molecules or modulating the function of IFN-y could offer a promising treatment alternative, especially for high-grade MCT. Alongside the previously mentioned checkpoint inhibitors, such therapies could be considered to enhance treatment outcomes.
Conclusion - Re-phrase the sentences for improving the scientific communicability.
Overall, it is indicated by our findings that high-grade MCT is associated with an immunosuppressive microenvironment that exhibits elevated RANK/RANK-L signaling and enhanced immune checkpoint immunoreactivity, potentially facilitating intratumorally immune escape. These biomarkers hold promise as clinically relevant indicators of disease progression and response to immunotherapy in dogs with high-grade MCT, emphasizing their importance for guiding treatment decisions and improving patient outcomes.

Round 2
Reviewer 2 Report
Dear Authors, the article is fine, but two refinements are necessary to define it ready for publication.
Line 189 change analysis with investigations;
In Table 4 include the word "regional" between skin location - skin regional location.
Author Response
Good afternoon:
Line 189 change analysis with investigations
Answer: correction performed as suggested.
In Table 4 include the word "regional" between skin location - skin regional location.
Answer: correction performed as suggested.
